# A Norbornadiene-Based Molecular System for the Storage of Solar–Thermal Energy in an Aqueous Solution: Study of the Heat-Release Process Triggered by a Co(II)-Complex

**DOI:** 10.3390/molecules28217270

**Published:** 2023-10-25

**Authors:** Franco Castro, Jorge S. Gancheff, Juan C. Ramos, Gustavo Seoane, Carla Bazzicalupi, Antonio Bianchi, Francesca Ridi, Matteo Savastano

**Affiliations:** 1Área Química Inorgánica, Departamento Estrella Campos, Facultad de Química, Universidad de la República, Gral. Flores 2124, Montevideo 11800, Uruguay; fcastro@fq.edu.uy; 2Laboratorio de Química Fina, Instituto Polo Tecnológico de Pando, Facultad de Química, Universidad de la República, By Pass de Rutas 8 y 101 s/n, Pando 91000, Uruguay; jcramos@fq.edu.uy; 3Graduate Program in Chemistry, Facultad de Química, Universidad de la República, Montevideo 11800, Uruguay; 4Área Química Orgánica, Departamento de Química Orgánica, Facultad de Química, Universidad de la República, Av. Gral. Flores 2124, Montevideo 11800, Uruguay; gseoane@fq.edu.uy; 5Department of Chemistry “Ugo Schiff”, University of Florence, Via della Lastruccia, 3-13, 50019 Sesto Fiorentino, Italy; carla.bazzicalupi@unifi.it (C.B.); antonio.bianchi@unifi.it (A.B.); francesca.ridi@unifi.it (F.R.); 6CSGI Consortium, Via della Lastruccia, 3, 50019 Sesto Fiorentino, Italy; 7Dipartimento di Scienze Umane e Promozione della Qualità della Vita, Università Telematica San Raffaele Roma, Via di Val Cannuta 247, 00166 Roma, Italy

**Keywords:** quadricyclane, norbornadiene, energy storage, solar energy, MOST

## Abstract

It is urgent yet challenging to develop new environmentally friendly and cost-effective sources of energy. Molecular solar thermal (MOST) systems for energy capture and storage are a promising option. With this in mind, we have prepared a new water-soluble (pH > 6) norbornadiene derivative (HNBD1) whose MOST properties are reported here. HNBD1 shows a better matching to the solar spectrum compared to unmodified norbornadiene, with an onset absorbance of λ_onset_ = 364 nm. The corresponding quadricyclane photoisomer (HQC1) is quantitatively generated through the light irradiation of HNBD1. In an alkaline aqueous solution, the MOST system consists of the NBD1^−^/QC1^−^ pair of deprotonated species. QC1^−^ is very stable toward thermal back-conversion to NBD1^−^; it is absolutely stable at 298 K for three months and shows a marked resistance to temperature increase (half-life *t*_½_ = 587 h at 371 K). Yet, it rapidly (*t*_½_ = 11 min) releases the stored energy in the presence of the Co(II) porphyrin catalyst Co-TPPC (Δ*H*_storage_ = 65(2) kJ∙mol^−1^). Under the explored conditions, Co-TPPC maintains its catalytic activity for at least 200 turnovers. These results are very promising for the creation of MOST systems that work in water, a very interesting solvent for environmental sustainability, and offer a strong incentive to continue research towards this goal.

## 1. Introduction

Finding efficient and sustainable alternatives to the use of fossil fuels for energy production is now an inescapable mission and solar energy offers numerous options [1,2]. The Sun is an ever-burning and very powerful energy source that radiates much more energy to the Earth than is currently needed to meet global needs [1]. Various technologies for the exploitation of solar energy are already available. Their application often depends on elements of peculiarity, including geographical and environmental circumstances [2]. In this context, the capture and storage of solar energy at the molecular level is a hot topic and molecules capable of absorbing light giving rise to stable photoisomers capable of releasing the stored energy on demand in the form of heat, which are referred to as molecular solar thermal (MOST) systems for energy storage, are now in the spotlight as they can store significant amounts of energy in a relatively small space [3,4,5,6,7,8,9,10,11,12,13,14,15,16,17].

Promising molecular candidates for MOST are norbornadiene (NBD) derivatives, which photoconvert to their quadricyclane (QC) counterparts (Figure 1a) with the potential to achieve long-term stable high energy storage density [5,13]. The unsubstituted NBD molecule is not of practical use, mainly due to its poor match to the solar spectrum and the low quantum yield [18,19], but improved NBD molecules able to absorb visible light can be constructed by inserting groups of different electron donating/accepting properties on one of its double bonds [4,8], thus obtaining a so-called “push-pull” conjugated system [20]. Yet, the insertion of substituents (especially bulky ones) can have deleterious effects on the energy storage density, as a consequence of increasing the molar mass of the MOST, and may not favor quantum yield [4]. Nevertheless, successful NBD/QC systems have been engineered and some test devices for solar energy capture and heat release have opened great expectations for future applications [21,22,23,24,25]. A schematic representation of a MOST-based plant for domestic use is shown in Figure 1b.

A molecular system must meet some requirements to have MOST potential: (1) the parent molecule (the NBD, in our case) should absorb a significant part of the energy of the solar spectrum; (2) the photoconversion should proceed with a high quantum yield, equal or close to one; (3) the photoisomer (the QC, in our case) should be stable over time; (4) the photoisomer and its parent molecule should absorb light in different regions; (5) the isomeric interconversion should be repeatable for many cycles (high cyclability); (6) energy should be stored for significant periods of time; (7) the back-conversion of the photoisomer (QC to NBD, in our case) should be exergonic but with high activation energy (Figure 1c); and (8) the molar weight of the parent molecule should be small, in order to achieve significant stored energy densities (per mass unit) [4,5,21].

The high activation energy of back-conversion (Δ*E_a_*, Figure 1) is an important requirement for long-term energy storage, i.e., to avoid the spontaneous (thermal) reversion of QC to the parent NBD form. Equally important is having an efficient method to lower this activation energy and recover the stored energy on demand. For this purpose, both electrochemical [26,27,28,29,30,31] and metal-based catalytic [3,25,32,33,34,35] methods have shown great potential. Focusing on catalytic methods, the use of heterogeneous catalysts appears to be a viable solution for realizing energy plants based on recycling a MOST system and operating continuously (as schematically represented in Figure 1b) [24,25,36,37,38,39,40,41,42]. Various metal complexes anchored on solid supports have proved useful to this end [24,25,35,40,41,42].

Some mechanisms for these catalytic processes promoted by metal complexes have been proposed based on both reactivity data [43,44] and theoretical calculations [24], all highlighting the active participation of the metal centers. According to a mechanism computed for the conversion of a QC to the parent NBD in the presence of a Co(II) phthalocyanines complex, one of the labile C–C bonds of QC gives rise to an oxidative addition to the metal center followed by the formation of a transient carbocation and successive regeneration of the double C=C bonds of NBD [24].

Almost all of the MOST systems tested so far have been designed to work in organic solvents and only a very few to be used in water [34,35,45,46,47,48,49,50], although the aqueous medium should be preferable for environmental sustainability. This is probably due to early observations of possible QC reactivity with water in the absence and in the presence of metal catalysts [45,51,52,53], despite stable and successful MOST functioning in water also being obtained [34,35,45,46,47,48,49,50].

We recently returned to address this topic with the study of a new MOST system in water. Results confirm that it is possible to prepare efficient MOST systems which are stable in aqueous media and can accumulate a considerable energy density through exposure to solar light, store it for long times and deliver it on demand using a metal-based catalyst. The new MOST system (HNBD1/HQC1) considered here is shown in Figure 2, together with the catalyst used, the Co(II) complex of 5,10,15,20-tetrakis(*p*-carboxyphenyl)porphyrin (Co-TPPC), which has already proved to be an effective catalyst for similar reactions [34,35,47]. This MOST system was chosen considering two main requirements of the parent NBD1 molecules. The first one is its solubility both in water and in organic solvents, hence the presence of the carboxyl and the pentylamide groups which have the additional effect of generating a push–pull conjugated system. Although this study concerns only the aqueous medium (alkaline), we did not want to overlook the possibility of extending this MOST to organic solvents. In alkaline aqueous solution, where HNBD1 and HQC1 are very soluble as deprotonated NBD1^−^ and QC1^−^ forms, this MOST consists of the NBD1^−^/QC1^−^ pair. The second one concerns economic considerations, which are a crucial aspect for the effective implementation of the MOST concept, trying to prepare a low cost and scalable NBD, hence the easy synthesis and use of cheap components, some of which are from-the-shelf compounds, such as pentylamine, which is easily available at a low cost thanks to its wide industrial use.

On the other hand, the Co-TPPC catalyst was chosen not only for its above-mentioned efficiency in both water and organic solvents but also for its demonstrated ability to be adsorbed onto activated carbon and function as a heterogeneous catalyst [35], which is much easier to implement in a recirculating set-up of energy absorption, storage and release, like the one shown in Figure 1b, and prospectively represents a continuation of this work.

In the present paper, however, we preliminarily focus on the homogeneous Co-TPPC catalyst in an aqueous solution. We report the synthesis and characterization of the new HNBD1 and related HQC1 molecules and the photophysical, thermodynamic and kinetic properties of the isomerization processes in which they, and their deprotonated forms (NBD1^−^, QC1^−^), are involved and which define their MOST behavior in water.

## 2. Results and Discussion

### 2.1. Acid Dissociation of HNBD1 and HQC1 and Solution Stability

HNBD1 and HQC1 are sparingly soluble in an acidic aqueous solution but are soluble in alkaline media where their carboxylic groups are deprotonated. Nevertheless, they are soluble and stable enough (see below) to be studied by means of UV-Vis absorption spectroscopy over all the pH range. The analysis of UV-Vis spectra (Figure 3) recorded for both isomers in the pH range 2–11 (298 K, 0.1 M NaCl) allowed the determination of their p*K*_a_ (see the experimental section): p*K*_a_ = 3.73(1) for HNBD1, p*K*_a_ = 3.60(1) for HQC1. According to these data, NBD1 and QC1 are completely deprotonated above pH 6 (Appendix A), allowing the good solubility of both compounds.

The stability of the quadricyclane photoisomer, concerning its thermal back-conversion to the norbornadiene form, and with respect to possible reactions with water, is a prerequisite of major importance for the long-term energy storage and appropriate functioning of the MOST system. The stability of deprotonated HQC1 (QC1^−^) in water, under rather aggressive pH conditions (pH 11) and aerobic atmosphere, was verified by ^1^H-NMR measurements performed at 298 K and 371 K; QC1^−^ was demonstrated to be very stable, as no alterations were observed during three months at 298 K and only a very slow interconversion to NBD1^−^ (accompanied by some degradation) occurred at 370 K with a half-life *t*_½_ = 587 h (Appendix A). Then, the thermal back-conversion of QC1^−^ to NBD1^−^ meets the requirement of high activation energy and the captured solar energy can be stored in water at ambient temperatures or higher for very long times.

### 2.2. Photochemical Properties

HNBD1/NBD1^−^ and HQC1/QC1^−^ satisfy further prerequisites necessary for their application as MOST systems which consist in the absorption of light in different regions by the parent molecule and its photoisomer and in the better matching of the solar spectrum by HNBD1 and NBD1^−^ (Figure 3). Furthermore, the combination of the donor and acceptor properties of the substituents inserted on one of the norbornadiene double bonds resulted in a better matching of the solar spectrum compared to the unsubstituted NBD (λ_onset_ at ~267 nm), shifting the onset of absorption up to 343 nm (absorption onset defined as log ε = 2) at pH 6–11 where the anionic NBD1^−^ species is formed. On lowering the solution pH, the λ_onset_ increases up to 364 nm with the formation of the neutral HNBD1 species (Figure 4). Despite this, we obtained only a marginal overlap with the solar spectrum, while, for a profitable MOST, it would be desirable to reach the visible region of the spectrum.

Above pH 6, the complete isomerization of NBD1^−^ to QC1^−^ is achieved upon irradiation at 275 nm, as shown by UV-Vis and ^1^H-NMR spectra (Appendix A). The photoisomerization process is very efficient; the relevant quantum yield Φ_iso_ = 71%, determined for the conversion of NBD1^−^ into QC1^−^ (Appendix A), indicates that the majority of all absorbed photons results in successful photoisomerization events. This is another prerequisite for an effective MOST which is fulfilled by HNBD1/NBD1^−^.

### 2.3. Back-Conversion of HQC1/QC1^−^ to HNBD1/NBD1^−^

We first attempted to determine the energy stored (Δ*H*_storage_) by HQC1 by means of differential scanning calorimetry (DSC). The recorded thermogram (Figure 5) showed a sequence of thermal effects spanning a large temperature range (50–150 °C). The first one is smooth and endothermic, it occurs in the 50–80 °C range and ends with a marked endothermic peak corresponding to the melting of HQC1 (T_melting_ = 82–85 °C). HQC1 melting, however, overlaps with an exothermic process extending up to 150 °C, which is attributed to the HQC1 interconversion to HNBD1. The whole process was visually followed with a Melting Point B-540 BÜCHI instrument (BÜCHI, Milan, Italy) with a rate of temperature increase of 2 °C∙min^−1^, as in DSC measurements, and the only phenomenon we could observe was melting of HQC1 at 82–85 °C; above this temperature the sample remained molten up to 150 °C. To complicate the thermal picture shown by the thermogram in Figure 5, the melting of HNBD1 (110–112 °C), which was studied using independent DSC analysis, is followed and partially overlapped by an endothermic process extending up to 150 °C (Appendix A), which is not visible in the thermogram of Figure 5 because it is probably covered by the strongly exothermic interconversion of HQC1 to HNBD1. Even assuming that there are no other hidden processes, it is not possible to extract a reliable value of Δ*H*_storage_ from the thermogram in Figure 5.

Since the central point of this study concerns the catalytically activated functioning of this MOST in water, we firstly studied the kinetics of the catalyzed conversion of QC1^−^ to NBD1^−^ in water by means of ^1^H-NMR measurements. These showed that the long-term stable aqueous solutions of QC1^−^ (298 K, pH 11) undergo rapid interconversion with the formation of the parent isomer (NBD1^−^) upon the addition of catalytic amounts of Co-TPPC, which is a good property of a profitable MOST system. The kinetic decay of QC1^−^ was followed at 298.0 ± 0.2 K (Figure 6) and the first-order rate constant (*k* = 1.05 × 10^−3^ s^−1^) and the half-life (*t*_½_ = 11 min) of the QC1^−^ to NBD1^−^ back-conversion could be determined. Furthermore, as shown by the ^1^H-NMR spectra, NBD1^−^ is completely recovered and no other compounds are generated during this catalytically induced back-conversion.

The energy recovered from QC1^−^ in water using the selected Co(II)-porphyrin catalyst Co-TPPC was measured by means of isothermal titration calorimetry (ITC). The ITC experiments showed that, upon the addition of the catalyst to solutions of QC1^−^ at pH 11 (298 K) (see the experimental section), an exothermic process is rapidly generated (Appendix A) corresponding to an enthalpy change Δ*H* = −65(2) kJ∙mol^−1^, which is consistent with the heat evolved by similar MOST systems catalytically sensitized in water [35,47]. The corresponding energy density is 0.26 MJ∙kg^−1^, which is a good value very close to the 0.3 MJ∙kg^−1^ limit indicated for a MOST of practical use [21,54]. Further ITC measurements were performed, by adding successive amounts of a QC1^−^ solution to the catalyst at pH 11 (see Section 3.3.2), in order to verify the maintenance of the catalytic activity of Co-TPPC under the adopted condition. No loss of activity was observed during the 200 catalytic turnovers we followed.

## 3. Materials and Methods

### 3.1. General

Unless otherwise noted, all reagents and solvents were purchased from commercial suppliers and used without further purification. Flash column chromatography was carried out with Silica gel 60 (J.T. Baker, 40 mm average particle diameter). Solvent proportions in mixtures are expressed as v:v ratios. All reactions and chromatographic separations were monitored using TLC conducted on 0.25 mm silica gel plastic sheets (Macherey/Nagel, Polygram, SIL G/UV254). TLC plates were visualized with UV light (254 nm).

### 3.2. Synthesis of HNBD1 and HQC1

HNBD1 was synthesized through a scalable four-step synthetic pathway (Figure 7). Protocols to prepare **1**–**3** were taken from reported procedures [55,56,57]. The synthetic procedures used are not enantioselective, so the compounds **2**, **3**, HNBD1 and HQC1 were obtained as racemates.

#### 3.2.1. Synthesis of Diethyl Bicyclo[2.2.1]hepta-2,5-diene-2,3-dicarboxylate (**1**)

A round-bottom flask was charged with diethylacetylene dicarboxylate (5.10 g, 30.0 mmol) and placed in an ice bath. Freshly distilled cyclopentadiene (3.05 g, 46.1 mmol) was added. After 15 min, the ice bath was removed and the mixture was allowed to react at ambient temperature for 24 h. The mixture was separated using flash column chromatography (gradient elution of hexanes to hexanes:ethyl acetate, 8:2) to obtain (**1**) as a colorless oil (5.10 g, 72%). ^1^H-NMR and ^13^C-NMR signals were in good agreement with the literature [55]. ^1^H-NMR (400 MHz, CDCl_3_): δ = 1.30 (t, *J* = 7.1 Hz, 6H), 2.08 (dt, *J* = 6.8, 1.6 Hz, 1H), 2.28 (dt, *J* = 6.8, 1.6 Hz, 1H), 3.93 (m, 2H), 4.23 (q, *J* = 7.1 Hz, 4H), 6.92 (s, 2H). ^13^C-NMR (100 MHz, CDCl_3_): δ = 14.3, 53.6, 61.1, 73.0, 142.6, 152.3, 165.4.

#### 3.2.2. Synthesis of 3-(ethoxycarbonyl)bicyclo[2.2.1]hepta-2,5-diene-2-carboxylic acid (**2**)

Compound (**1**) (2.00 g, 8.47 mmol) was dissolved in 50 mL of a 1:1 mixture of EtOH:THF in a round-bottom flask. To the stirring solution, 80 mL of an aqueous solution of KOH (1%) was added. The mixture was stirred at 50 °C for 2 h until completion. The flask was placed in an ice bath and the mixture was acidified with concentrated HCl until pH = 1. This mixture was extracted using ethyl acetate (3 × 150 mL). The combined extracts were washed with water (2 × 60 mL) and brine (100 mL). The organic phase was dried over Na_2_SO_4_ and filtered. The solvent was removed under reduced pressure and the crude was purified using flash column chromatography (gradient elution of hexanes:ethyl acetate, 9:1 to hexanes:ethyl acetate, 1:1) to obtain (**2**) as a white powder (1.70 g, 96%). ^1^H-NMR and ^13^C-NMR signals were in good agreement with the literature [56]. ^1^H-NMR (400 MHz, CDCl_3_): δ = 1.38 (t, *J* = 7.1 Hz, 3H), 2.10 (dt, *J* = 7.2, 1.5 Hz, 1H), 2.21 (dt, *J* = 7.1, 1.7 Hz, 1H), 4.08 (s, 1H), 4.20 (s, 1H), 4.35 (m, 2H), 6.87 (dd, *J* = 5.0, 3.1 Hz, 1H), 6.90 (dd, *J* = 4.9, 3.3 Hz, 1H), 10.88 (s, 1H). ^13^C-NMR (100 MHz, CDCl_3_): δ = 14.0, 53.4, 54.8, 63.3, 72.7, 141.9, 142.8, 151.1, 162.3, 163.3, 167.7.

#### 3.2.3. Synthesis of Ethyl 3-(pentylcarbamoyl)bicyclo[2.2.1]hepta-2,5-diene-2-carboxylate (**3**)

Compound (**3**) was synthesized following a reported protocol [57]. To a stirred solution of (**2**) (1.00 g, 4.81 mmol) in CHCl_3_ (50 mL) was added NEt_3_ (3.36 mL, 24.1 mmol), HBTU (2.55 g, 6.72 mmol) and amylamine (0.78 mL, 6.73 mmol). The mixture was refluxed for 1 h. Once cooled, the mixture was washed with saturated solutions of NaHCO_3_ (40 mL), NH_4_OH (2 × 40 mL) and NH_4_Cl (40 mL), HCl 1.5% (20 mL), water (10 mL) and saturated brine (50 mL). The organic extract was dried over Na_2_SO_4_ and filtered. The solvent was removed under reduced pressure and the crude was purified using flash column chromatography (hexanes: ethyl acetate, 8:2) to obtain (**3**) as a colorless oil (1.19 g, 89%). ^1^H-NMR (Appendix A, 400 MHz, CDCl_3_): δ = 0.85 (m, 3H), 1.31 (m, 7H), 1.52 (m, 2H), 1.94 (dt, *J* = 7.0, 1.6 Hz, 1H), 2.07 (dt, *J* = 6.9, 1.6 Hz, 1H), 3.26 (m, 2H), 4.00 (s, 1H), 4.21 (m, 2H), 4.24 (s, 1H), 6.83 (m, 1H), 6.87 (m, 1H), 9.12 (s, 1H). ^13^C-NMR (Appendix A, 100 MHz, CDCl_3_): δ = 14.0, 14.2, 22.4, 29.0, 29.2, 39.8, 54.0, 54.7, 61.5, 71.0, 142.1, 142.8, 146.3, 162.7, 162.9, 166.3. HRMS calculated for C_16_H_23_NO_3_ ([M+H]^+^): 278.1756; exp 278.1760.

#### 3.2.4. Synthesis of 3-(pentylcarbamoyl)bicyclo[2.2.1]hepta-2,5-diene-2-carboxylic Acid (HNBD1)

Compound (**3**) (1.20 g, 4.33 mmol) was dissolved in 32 mL of a 1:1 mixture of EtOH:THF in a round-bottom flask. To the stirring solution, 30 mL of a 10% KOH aqueous solution was added. The mixture was stirred at 50 °C for 30 min until completion. The flask was placed in an ice bath and the mixture was acidified with concentrated HCl until pH = 1 and extracted with ethyl acetate (3 × 100 mL). The combined extracts were washed with water (2 × 30 mL) and brine (100 mL). The organic phase was dried over Na_2_SO_4_ and filtered. The solvent was removed under reduced pressure and HNBD1 was obtained as a pure, white powder (0.96 g, 89%). ^1^H-NMR (Appendix A, 400 MHz, CDCl_3_): δ = 0.89 (t, *J* = 6.9, 3H), 1.33 (m, 4H), 1.62 (m, 2H), 2.12 (dt, *J* = 7.0, 1.6 Hz, 1H), 2.23 (dt, *J* = 7.0, 1.7 Hz, 1H), 3.39 (m, 2H), 3.94 (s, 1H), 4.21 (s, 1H), 6.87 (dd, *J* = 5.0, 3.0 Hz, 1H), 6.96 (dd, *J* = 5.0, 3.2 Hz, 1H), 7.17 (s, 1H), ^13^C-NMR (Appendix A, 100 MHz, CDCl_3_): δ = 14.1, 22.4, 28.8, 29.2, 40.7, 53.3, 54.6, 72.2, 140.1, 143.8, 153.0, 158.1, 164.9, 165.5. HRMS calculated for C14H19NO3 ([M+H]^+^): 250.1443; exp 250.1448. M.p. = 110–112 °C. Anal. Calcd. for C_14_H_19_NO_3_: C, 67.45; H, 7.68; N, 5.62. Found: C, 67.32; H, 7.74; N, 5.54.

#### 3.2.5. Synthesis of 5-(pentylcarbamoyl)tetracyclo[3.2.0.0^2,7^.0^4,6^]heptane-1-carboxylic Acid (HQC1)

In a quartz cuvette, HNBD1 (350 mg) was dissolved in MeCN (4 mL) and purged with argon for 10 min. The cuvette was placed inside a custom-made aluminum box equipped with an LED light source with a wavelength of 275 nm, connected to a power source. The LED was switched on and the cuvette was irradiated for 24 h. The solvent was removed under reduced pressure and the crude was purified using flash column chromatography (gradient elution of hexanes:ethyl acetate 6:4 to hexanes:ethyl acetate 4:6) to separate the QC from the unconverted NBD. HQC1 was obtained as a white solid (185 mg, 53%). ^1^H-NMR (Appendix A, 400 MHz, CDCl_3_): δ = 0.88 (t, *J* = 6.8 Hz, 1H), 1.28 (m, 4H), 1.49 (p, *J* = 7.3 Hz, 2H), 2.13 (dt, *J* = 11.9, 1.4 Hz, 1H), 2.29 (dt, *J* = 11.9, 1.4 Hz, 1H), 2.36 (dd, *J* = 4.9, 1.6 Hz, 1H), 2.45 (dd, *J* = 4.9, 1.6 Hz, 1H), 2.49 (dd, *J* = 5.0, 2.4 Hz, 1H), 2.63 (dd, *J* = 4.9, 2.4 Hz, 1H), 3.19 (td, *J* = 7.3, 5.5 Hz, 2H). ^13^C-NMR (Appendix A, 100 MHz, CDCl_3_): δ = 175.6, 172.3, 40.0, 37.9, 35.1, 30.9, 30.7, 30.5, 29.2, 29.1, 27.7, 22.4, 14.1. HRMS calculated for C14H19NO3 ([M+H]^+^): 250.1443; exp 250.1431. M.p. 82–85 °C. Anal. Calcd. for C_14_H_19_NO_3_: C, 67.45; H, 7.68; N, 5.62. Found: C, 67.39; H, 7.71; N, 5.57.

### 3.3. Physicochemical Properties of HNBD1/NBD1^−^ and HQC1/QC1^−^

The physicochemical properties determined for HNBD1/NBD1^−^ and HQC1/QC1^−^ are collected in Table 1.

#### 3.3.1. Spectrophotometric Measurements

UV-Vis absorption spectra were recorded at 298 K using either a Jasco V-670 spectrophotometer (Jasco Europe, Lecco, Italy) or a Shimadzu UV-1900i spectrophotometer (Shimadzu, Montevideo, Uruguay). Photoisomerization experiments were carried out using 275 nm LED lamps (Seoul Viosys, AAP series, CUD8AF4D) (Seoul Viosys, Bradford, UK). The LED was set to 50% of the relative radiant flux for UV and quantum yield measurements, and to 90% for ^1^H-NMR experiments. In a quartz cuvette with a magnetic stirrer, 3 mL of a 152 µM H_2_O solution of NBD1^−^ at pH = 10.96 was irradiated under a 275 nm LED and the UV-Vis spectra were recorded periodically (Appendix A). Measurements performed to determine the p*K*_a_ values of HNBD1 and HQC1 were performed in 0.1 NaCl solution at 298.1 K with solutions 1.1 × 10^−4^ M in the pH range 2–11. The pH was varied through addition of measured amounts of standardized NaOH or HCl solutions. The equilibrium constants for NBD1 and QC1 deprotonation were determined by treating these spectra with the computer program HypSpec [58].

#### 3.3.2. Isothermal Titration Calorimetry (ITC)

The heat released in the catalytic back-conversion of QC1^−^ to NBD1^−^ was measured in aqueous solution at pH 11 and 298.1 K by using a TAM III microcalorimeter (TA Instrument, Waters, Milan, Italy) and a procedure already described [59]. The microcalorimeter was checked by determining the enthalpy of reaction of strong base (NaOH) with strong acid (HCl) solutions. The value obtained (−56.7(2) kJ∙mol^−1^) was in agreement with the values in the literature [60]. Two different experiments were performed to catalyze back-conversion. In the first one, about 1.2 cm^3^ of a 3.32 mM QC1^−^ solution was charged into the calorimetric cell and allowed to reach thermal equilibration. Then, about 10 μL of an aqueous solution of Co-TPPC at pH 11 was added in such a way as to have the catalyst at about 5% with respect to QC1^−^. The thermal effect was recorded and corrected for the heats of dilution determined in separate experiments. The heat released in the catalytic back-conversion was calculated considering 100% conversion of QC^−^ to NBD^−^ according to NMR measurements. Seven independent measurements obtained with two different batches of the catalyst were performed to obtain the final value. In the second experiment, the calorimetric cell was charged with about 1.0 cm^3^ of a solution of Co-TPPC at pH 11 and, once the thermal equilibration was reached, 10 additions of 100 μL each of a 0.100 M QC1^−^ solution at pH 11 were added. We waited 180 min between one addition and the next to ensure recovery of the thermogram baseline. Also in this case, the thermal effects were recorded and corrected for the heats of dilution determined in a separate experiment. The heat released in each of the successive catalytic back-conversions was calculated considering the 100% conversion of QC^−^ to NBD^−^ according to NMR measurements. The value obtained in the second experiment was equal within the experimental errors to the value obtained in the first one (Δ*H*_storage_ = 65(2) kJ∙mol^−1^) and no loss of catalytic activity was observed during the 10 interconversion processes followed in the second experiment corresponding to a catalyst turnover number of 200.

#### 3.3.3. Differential Scanning Calorimetry (DSC)

Differential scanning calorimetry (DSC) measurements were performed by means of a Discovery DSC 2500 from TA Instruments (TA Instrument, Waters, Milan, Italy). Approximately 2 mg of HQC1 was placed in a Tzero Aluminum Hermetic pan. The HQC1 to HNBD1 transformation was followed using the following temperature program: ramp from 20 °C to 170 °C at 2 °C∙min^−1^.

#### 3.3.4. Nuclear Magnetic Resonance (NMR)

NMR spectra were recorded using either Neo 400 Bruker Advance (400 MHz for ^1^H-NMR and 100 MHz for ^13^C-NMR in CDCl_3_ or D_2_O) or Bruker Advance 400 (400 MHz for ^1^H in D_2_O) spectrometers (Bruker, Billerica, MA, USA). Proton chemical shifts were reported in ppm using residual solvent as an internal reference (CDCl_3_, 7.26 ppm; D_2_O, 4.79 ppm), and carbon chemical shifts were reported in ppm relative to the center-line of the CDCl_3_ triplet (77.0 ppm). The solution pH was adjusted by using D_2_O solutions of DCl and NaOD. The pH of the solution was calculated from the measured pD value by using the relationship pH = 0.929 × pD + 0.41 [61]. For the NBD1^−^ to QC1^−^ interconversion measurements, 5 mg (0.02 mmol) of HNBD1 was dissolved in minimal amounts of a dilute solution of NaOD, and the pD was adjusted using a dilute D_2_O solution of DCl and NaOD. The final volume was brought to 1 mL with D_2_O, and the final pD was measured before irradiation (pD = 11.04, pH = 10.67). The solution was placed in an NMR tube and the UV LED was shone into the NMR tube. The experiments were carried out in a custom-made aluminum box that allowed the NMR tube to be placed at a fixed distance from the built-in LED and protected it from external light. The ^1^H-NMR spectra of the solution (Appendix A) were immediately measured after each successive irradiation period. In the catalyzed back-conversion measurements, the concentrations of QC1^−^ and Co-TCCP were 15.1 mM and 0.711 mM, respectively.

#### 3.3.5. Photoisomerization Quantum Yield

The quantum yield for the photoisomerization of NBD1^−^ to QC1^−^ was determined following a reported protocol based on a ferrioxalate actinometry [62]. The photoisomerization quantum yield (*ϕ*) was determined under a total absorption regime (absorbance of NBD1^−^ at 275 nm > 2). Absorbance values at 300 nm were used to calculate the NBD1^−^ concentration at each irradiation time. The experiments were carried out using a 1.524 mM solution of NBD1^−^ in water at a pH of 10.96 (Appendix A).

#### 3.3.6. Mass Spectrometry

HRMS was obtained using a Q Exactive Plus mass spectrometer (Thermo Scientific, Mundelein, IL, USA) using direct injection (5 μL∙min^−1^) using MeOH as solvent and an Ion Max API source with a HESI-II probe. The mass spectrometer was operated in a positive mode, ion spray voltage was set at 3.5 kV and capillary temperature at 250 °C.

## 4. Conclusions

The objective of the present work was the construction of a MOST system based on a pair of norbornadiene/quadricyclane photoisomers capable of functioning in water, considering the lower environmental impact that this solvent would have compared to organic solvents.

For this purpose, we have synthesized and characterized a scalable new water-soluble norbornadiene derivative (HNBD1) and tested its performance as a NBD1^−^/QC1^−^ MOST system in an alkaline aqueous solution. HNBD1 and NBD1^−^ show an improved, although not yet satisfactory, matching with the solar spectrum, in comparison to the unsubstituted NBD, with λ_onset_ of 364 nm and 343 nm at pH 2 and 11, respectively. Conversely, HQC1 and QC1^−^ absorb at shorter wavelengths. NBD1^−^ is photochemically converted to QC1^−^ in a quantitative manner and with high photoisomerization quantum yield (71%). The generated photoisomer (QC1^−^) is 100% stable toward the thermal back-conversion to the parent NBD1^−^ for at least three months at ambient temperature and has a long half-life (t_1/2_ = 587 h) even at high temperatures (371 K). A good energy storage density of 0.26 MJ∙kg^−1^ was determined in water by means of ITC following the fast back-conversion of QC1^−^ to NBD1^−^ sensitized by a Co(II) porphyrin catalyst (Co-TPPC), whose catalytic activity is maintained for at least 200 turnovers. However, for a productive application of the MOST concept, a better matching of the system to the solar spectrum, possibly achieving the photoconversion of norbornadiene in the visible region, must be pursued. These results form a basis for the design and development of more efficient systems in this highly environmentally sustainable solvent.

## Figures and Tables

**Figure 1 molecules-28-07270-f001:**
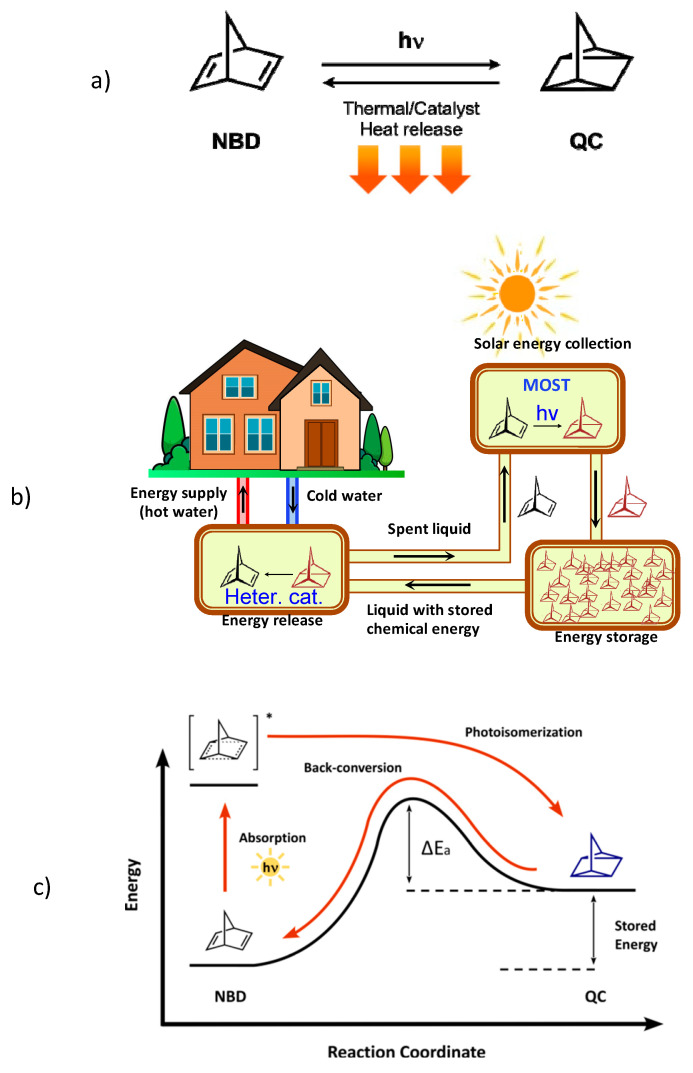
(**a**) NBD and QC structures. (**b**) Schematic representation of a MOST-based plant for domestic use. (**c**) Photochemical and thermodynamic functioning of MOST molecules.

**Figure 2 molecules-28-07270-f002:**
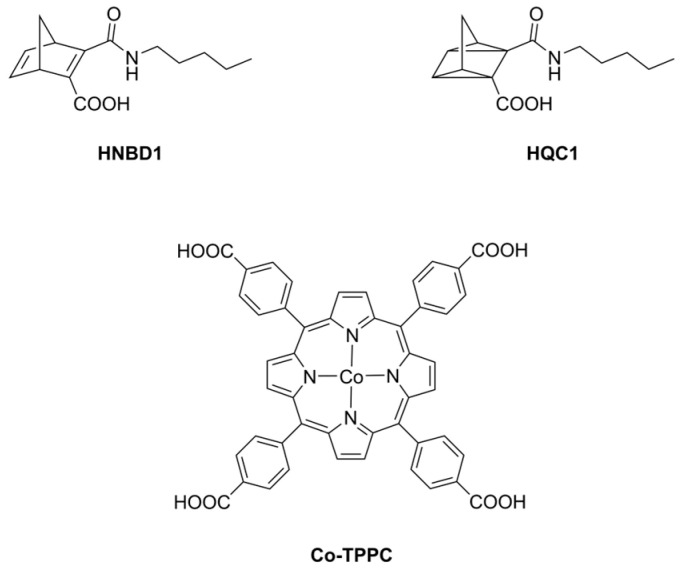
HNBD1, HQC1 and Co-TPPC structures.

**Figure 3 molecules-28-07270-f003:**
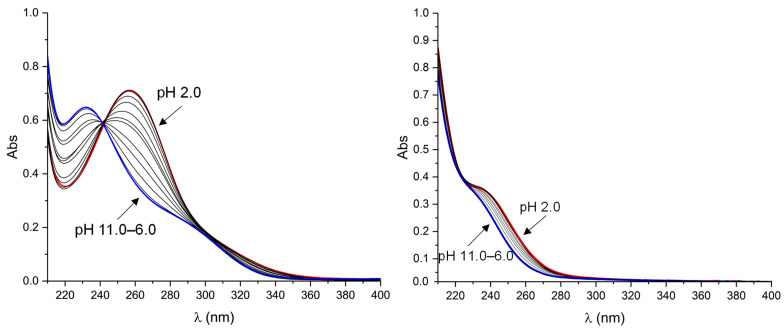
UV-Vis spectra of HNBD1 and HQC1 recorded at different pH values in the range 2.0–11.0. [HNBD1] = [HQC1] = 1.1 × 10^−4^ M; 298 K, 0.1 M NaCl; Spectra in the pH range 6.0–11.0 are identical. Color code: red, spectra recorded at pH 2; blue, spectra recorded in the pH range 6.0–11.0.

**Figure 4 molecules-28-07270-f004:**
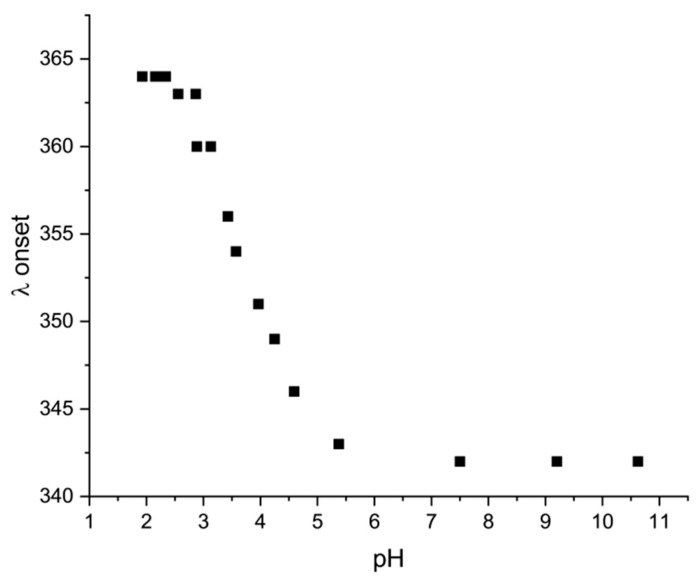
pH dependence of the λ_onset_ for HNBD1/NBD1^−^.

**Figure 5 molecules-28-07270-f005:**
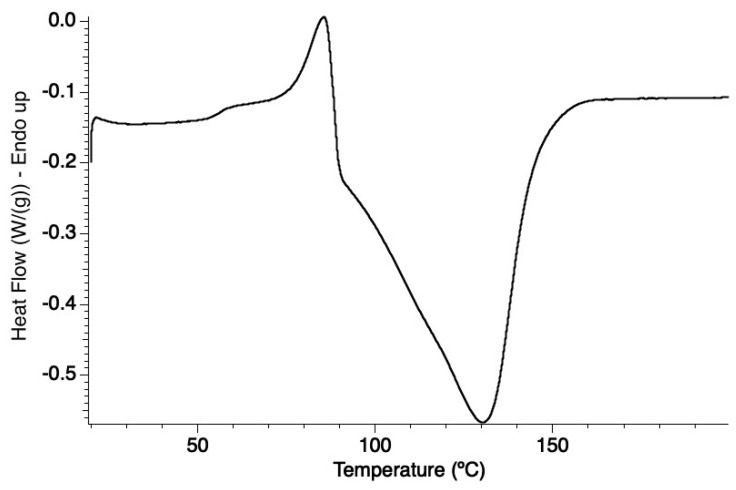
DSC thermogram recorded for HQC1 with a rate of temperature increase of 2 °C∙min^−1^.

**Figure 6 molecules-28-07270-f006:**
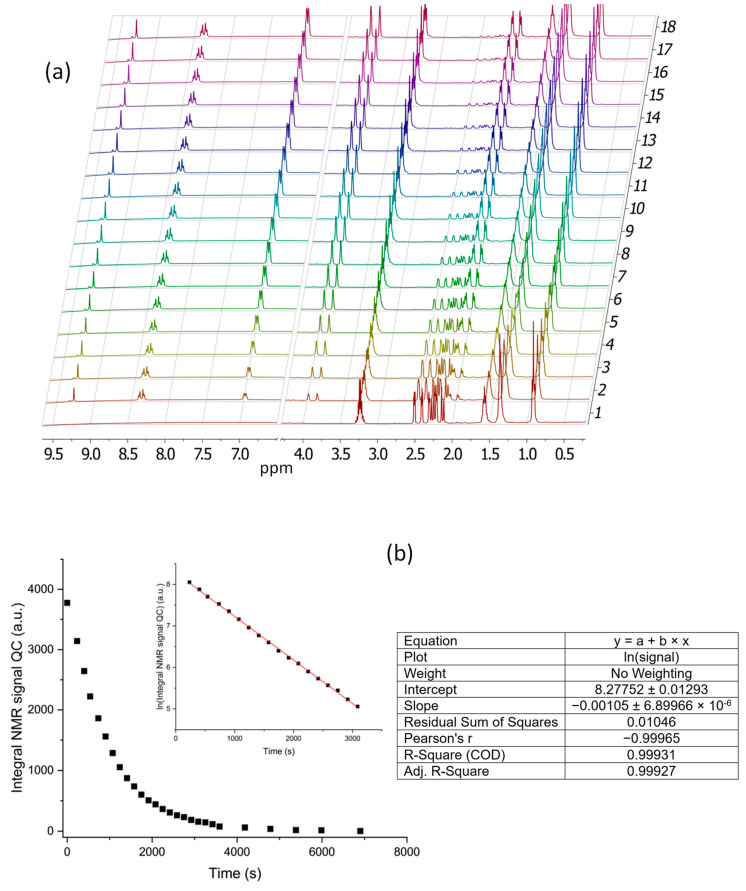
(**a**) ^1^H-NMR spectra (D_2_O, pH 11, 298 K, 400 MHz) showing the evolution with time of the catalyzed back-conversion of QC1^−^ to NBD1^−^. Spectrum 1 corresponds to QC1^−^ without catalysts, spectrum 2 was recorded after catalyst addition and mixing (230 s), and then the reaction was left to freely progress; spectrum 18 corresponds to 3086 s total time. (**b**) Left: variation of the integral of the ^1^H-NMR signal of QC1^−^ at 2.5 ppm with time and fitting of the corresponding logarithmic curve. Right: Fitting details and parameters.

**Figure 7 molecules-28-07270-f007:**
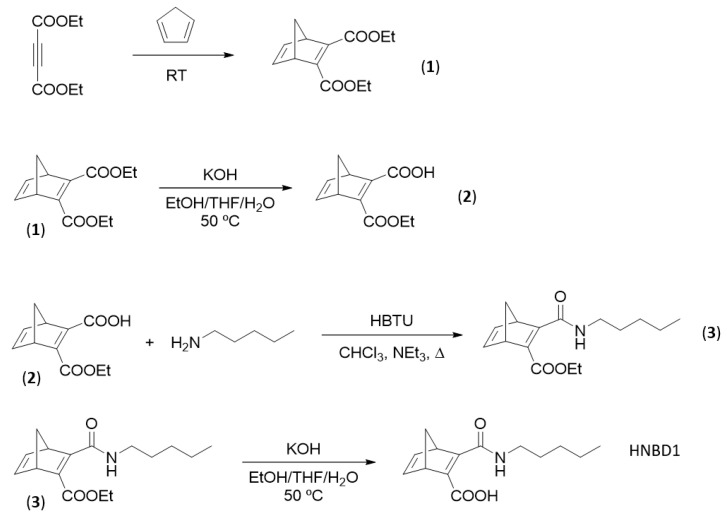
Schematic procedure for the synthesis of HNBD1.

**Table 1 molecules-28-07270-t001:** Physicochemical properties of HNBD1/NBD1^−^ and HQC1/QC1^−^.

HNBD1/NBD1^−^			
p*K*_a_ ^a^	λ_onset_ ^a^ (nm)	Φ_iso_ ^a^ (%)	Δ*H*_storage_ ^b^ (kJ∙mol^−1^)
3.73(1)	364 (pH 2); 343 (pH 11)	71	65(2)
HQC1/QC1^−^			
p*K*_a_ ^a^	*t*_½_^c^ Thermal, 298 K	*t*_½_^c^ Thermal, 371 K	*t*_½_^c^ Cat. Co-TPPC, 298 K
3.60(1)	100% stable over 3 months	587 h	11 min

(^a^) Measured in water at 298.1 ± 0.1 K. (^b^) Measured by means of ITC in water for the interconversion of HQC1 to HNBD1 in the presence of Co-TPPC (pH 11, 298.1 K). (^c^) Half-life of the thermal and catalyzed back-conversions of QC1^−^ to NBD1^−^ in water at pH 11. At 298.1 K in the presence of Co-TPPC.

## Data Availability

The data presented in this study are available on request from the authors.

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
