# Peer review of "A Norbornadiene-Based Molecular System for the Storage of Solar–Thermal Energy in an Aqueous Solution: Study of the Heat-Release Process Triggered by a Co(II)-Complex"

_molecules, 2023, doi:10.3390/molecules28217270_

Round 1

Reviewer 1 Report

I have read the contribution titled “A norbornadiene-based molecular system for the storage of solar-thermal energy in aqueous solution: study of the 3 heat-release process triggered by a Co(II)-complex” by Savastano and co-workers with much interest. What they are describing is a contribution to the development of chemical storage systems of solar energy.

The paper is well written and as far as I can judge it the execution of the synthetic work and also the analytical characterization have been carried out expertly.

A short literature search brought me to the conclusion that much of the work is based on previous work of a Japanese research group around Kazuhiro Maruyama in the 1980s. While some of these papers are referenced in the work I notice that others are not. Especially the

I would suggest to add the following references:

[1]          K. Maruyama, H. Tamiaki, S. Kawabata, J. Chem. Soc., Perkin Trans. 2 1986, 543–549.

[2]          K. Maruyama, K. Terada, Y. Yamamoto, Chem. Lett. 1981, 10, 839–842.

[3]          K. Maruyama, H. Tamiaki, S. Kawabata, Chem. Lett. 1984, 13, 743–746.

[4]          K. Maruyama, H. Tamiaki, Chem. Lett. 1986, 15, 819–822.

[5]          Y. Kawabata, H. Kumoyama, T. Imasaka, N. Ishibashi, Analyt. Chim. Acta 1991, 243, 97–101.

[6]          K. Maruyama, H. Tamiaki, S. Kawabata, J. Org. Chem. 1985, 50, 4742–4749.

I am really a bit concerned how the authors could have not referenced the J.Org. Chem paper which even in its title “Development of a solar energy storage process. Photoisomerization of a norbornadiene derivative to a quadricyclane derivative in an aqueous alkaline solution” shows it relevance to the current work.

In another reference related issue, I want to state that the sentence “…. Co(II) complex of  5,10,15,20-tetrakis(p-carboxyphenyl)porphyrin (Co-TPPC), which has already proved to be an effective catalyst for similar reactions.” requires of course references to examples where the effectivity of said complex was proven!

Overall this is a nice paper. I wish the authors would have had the courage to point out that most of the chemistry they are presenting is based on the work of the mentioned Japanese group, instead of clumsily hiding this fact from the reader.

After fixing the point mentioned above, I support publication of the article in Molecules.

English is quite good!

Author Response

Reviewer 1

I have read the contribution titled “A norbornadiene-based molecular system for the storage of solar-thermal energy in aqueous solution: study of the 3 heat-release process triggered by a Co(II)-complex” by Savastano and co-workers with much interest. What they are describing is a contribution to the development of chemical storage systems of solar energy.

The paper is well written and as far as I can judge it the execution of the synthetic work and also the analytical characterization have been carried out expertly.

Authors

Many thanks to this reviewer for her/his kind words and for appreciating our work.

Reviewer 1

A short literature search brought me to the conclusion that much of the work is based on previous work of a Japanese research group around Kazuhiro Maruyama in the 1980s. While some of these papers are referenced in the work I notice that others are not. Especially the

I would suggest to add the following references:   

[1]          K. Maruyama, H. Tamiaki, S. Kawabata, J. Chem. Soc., Perkin Trans. 2 1986, 543–549.

[2]          K. Maruyama, K. Terada, Y. Yamamoto, Chem. Lett. 198110, 839–842.

[3]          K. Maruyama, H. Tamiaki, S. Kawabata, Chem. Lett. 198413, 743–746.  

[4]          K. Maruyama, H. Tamiaki, Chem. Lett. 198615, 819–822.

[5]          Y. Kawabata, H. Kumoyama, T. Imasaka, N. Ishibashi, Analyt. Chim. Acta 1991243, 97–101.

[6]          K. Maruyama, H. Tamiaki, S. Kawabata, J. Org. Chem. 198550, 4742–4749.

I am really a bit concerned how the authors could have not referenced the J.Org. Chem paper which even in its title “Development of a solar energy storage process. Photoisomerization of a norbornadiene derivative to a quadricyclane derivative in an aqueous alkaline solution” shows it relevance to the current work.

Authors

In the original manuscript, we had already included 9 references to the Maruyama group's publications. One of them is the first in the list of additional references suggested by the reviewer. It seemed to us that the Maruyama group was sufficiently cited, in particular with regards to the study of MOSTs in water: it should be noted that of the 7 references used for this purpose, 6 refer to this group, highlighting that it is the group that started to study MOSTs in water. Nevertheless, we gladly added two more references (refs. 54 and 55, lines 545-549), the third and the sixth in the suggested list, which refer to the use of MOSTs in water.

Reviewer 1

In another reference related issue, I want to state that the sentence “…. Co(II) complex of  5,10,15,20-tetrakis(p-carboxyphenyl)porphyrin (Co-TPPC), which has already proved to be an effective catalyst for similar reactions.” requires of course references to examples where the effectivity of said complex was proven!

Authors

This sentence is now referenced (line 102).

Reviewer 1

Overall this is a nice paper. I wish the authors would have had the courage to point out that most of the chemistry they are presenting is based on the work of the mentioned Japanese group, instead of clumsily hiding this fact from the reader.

Authors

We wish to thank this reviewer for her/his restyled kind words. If we wanted to hide the merits of the Maruyama group, we would not have mentioned it as many times as we did from the beginning.

Reviewer 2 Report

The idea of the work is good and the topic is important. However, the reported work requires improvement and revisions.

1.       'Abstract' should more focus on main research outcomes and novelty should mention, which is missing. Add 1 or 2 lines as per novelty of work  in 'Abstract' section.

2. Some updated refs from HNBDs could be cited, such as Molecules 2023, 28, 4507; Org. Chem. Front., 2020,7, 3515-3520; J. Org. Chem. 2019, 84, 14627−14635 and Org. Lett. 2020, 22, 8086−8090

3. Paper must provide a comprehensive critical review of recent developments in a specific area or theme.

4. The authors should do the analysis the conclusion section must clearly establish a strong correlation with the proposed topic.

5. The objective or objectives should be clearly elucidated in the last paragraph of the introduction.

6. I suggest the also could list a Table for comparison the document on its Molecular solar thermal (MOST) systems.

7. The quality of Fig1 could be improved.

polish the full MS.

Author Response

Reviewer 2

The idea of the work is good and the topic is important. However, the reported work requires improvement and revisions.

Authors

Many thanks to this reviewer for he/his kind words and suggestions.

Reviewer 2

  1. 'Abstract' should more focus on main research outcomes and novelty should mention, which is missing. Add 1 or 2 lines as per novelty of work  in 'Abstract' section.

Authors

The abstract has been modified. We hope we have satisfied this reviewer's expectations.

Reviewer 2

  1. Some updated refs from HNBDs could be cited, such as Molecules 2023, 28, 4507; Org. Chem. Front., 2020,7, 3515-3520; J. Org. Chem. 2019, 84, 14627−14635 and Org. Lett. 2020, 22, 8086−8090

Authors

The suggested references are not relevant to the content of the manuscript and, for this reason, we have not included them. This was surely an inadvertent mistake. We will be happy to consider other references according to new reviewer's suggestions.

Reviewer 2

  1. Paper must provide a comprehensive critical review of recent developments in a specific area or theme.

Authors

A comprehensive critical review on recent developments in this area would produce enough material for a separate publication, given its vastness. It would hardly find space in the introduction of this paper. We believe that the 55 references already present in the introduction are sufficient to represent the subject area.

Reviewer 2

  1. The authors should do the analysis the conclusion section must clearly establish a strong correlation with the proposed topic.

Authors

The beginning of the conclusion section (lines 373-377) was modified to include the suggested correlation.

Reviewer 2

  1. The objective or objectives should be clearly elucidated in the last paragraph of the introduction.

Authors

We have modified the last part of the introduction (from line 115). We hope we have satisfied this reviewer's request.

Reviewer 2

  1. I suggest the also could list a Table for comparison the document on its Molecular solar thermal (MOST) systems.

Authors:

Doing this would be, more or less, like doing the comprehensive critical review previously proposed, with the complication of having to report in the table a multitude of information relating to the photophysical, thermochemical and thermodynamic properties by referring to the multitude of experimental conditions and methodologies used. We would probably need more than one table to do this. Such a data collection would also be a work in itself. We do not believe this is appropriate for our paper, a normal publication.

Reviewer 2

  1. The quality of Fig1 could be improved.

 Authors

The quality of the original Figure 1 is good. We will handle the matter with the editorial team to ensure that its final appearance is better than the current one.

Reviewer 3 Report

The paper submitted for review concerns a very important energy problem of our civilization - a fuller use of solar energy through its reversible accumulation in the potential energy of chemical compounds and its controlled release.

The Authors exploited the well-known fact of a light-induced electrocyclic reaction that converts a norbornadiene derivative into a "tetracyclate" containing two cyclopropane rings. Their work consisted in the synthesis of a stable derivative of norbornadiene, easy soluble in an aqueous alkaline environment, which also has an absorption band best suited to the energy source - sunlight. The release of the accumulated energy was planned using a porphyrin Co(II) complex.

I must emphasize that the planned goal was indeed ambitious, but I have considerable doubts about the results obtained, intensified by the fact that the material provided to me lacked Supplementary Materials, which the Authors repeatedly referred to.

The structure chosen by the Authors as optimal is the monoamide of norbornadiene dicarboxylic acid, but its absorption spectrum is not matched to the emission spectrum of the Sun. In a neutral solution, it has a maximum absorption located about 260 mn, and in an alkaline solution, recommended by the Authors for a higher concentration, it has a maximum absorption of just over 230 nm. Both of these values are at the limit of sunlight in the UVC region, which accounts for solar energy by well below 1%. Therefore, I do not consider the chemical modification to be a success that brings it closer to practical use.

The Authors have shown that the photoproduct called "tetracycrate" can be converted into a substrate in an exothermic reaction (approx. 66kJ/mol) with the help of a soluble Co(II) porphyrine complex, but they do not suggest how to remove this complex later before the substrate is photochemically reused. They did not conduct any studies on whether this immobilized complex is still active or not. Thus, the energy cycle suggested in the article has no evidence for the effectiveness of the concept. I understand that this is preliminary research and perhaps further studies will provide an efficient method of energy accumulation and recovery.

The Authors devoted a lot of space in their work to organic synthesis and here I also have comments and questions. In step 2 of gentle saponification, a solution containing THF was used, for what purpose? The same question applies to stage 4.

Also, How do you know which ester group has been saponified? I believe that each of the two had the same chances, so the product is probably an equimolar mixture of both possible monoesters, which unfortunately is not visible either in the reaction diagram or in the nomenclature used in the description (line 235), as well as in the description of step 3 (line 250).

Considering the methods of amide synthesis, why was the transformation process of (1) into (3) not performed in a single step by acting on the diester amine? One explanatory sentence would be useful here.

To sum up, I think that the work should be reworded, and the conclusions drawn from it should be more toned down. Once these changes have been made and the supplementary material has actually been made available, I believe it can be published.

Author Response

Reviewer 3

The paper submitted for review concerns a very important energy problem of our civilization - a fuller use of solar energy through its reversible accumulation in the potential energy of chemical compounds and its controlled release.

The Authors exploited the well-known fact of a light-induced electrocyclic reaction that converts a norbornadiene derivative into a "tetracyclate" containing two cyclopropane rings. Their work consisted in the synthesis of a stable derivative of norbornadiene, easy soluble in an aqueous alkaline environment, which also has an absorption band best suited to the energy source - sunlight. The release of the accumulated energy was planned using a porphyrin Co(II) complex.

I must emphasize that the planned goal was indeed ambitious, but I have considerable doubts about the results obtained, intensified by the fact that the material provided to me lacked Supplementary Materials, which the Authors repeatedly referred to.

Authors

We thank this reviewer very much for his comments and suggestions which gave us the possibility to improve the manuscript and supplement it with further results. We regret that she/he did not have the opportunity to view the supplementary material. We apologize to her/him for this. Due to our omission in the manuscript submission process, this material became available late. Surely she/he can see it now.

Reviewer 3

The structure chosen by the Authors as optimal is the monoamide of norbornadiene dicarboxylic acid, but its absorption spectrum is not matched to the emission spectrum of the Sun. In a neutral solution, it has a maximum absorption located about 260 mn, and in an alkaline solution, recommended by the Authors for a higher concentration, it has a maximum absorption of just over 230 nm. Both of these values are at the limit of sunlight in the UVC region, which accounts for solar energy by well below 1%. Therefore, I do not consider the chemical modification to be a success that brings it closer to practical use.

Authors

All this is absolutely true. We didn't try to hide it: by saying that our norbornadiene photoconverts with a Lambda-onset = 364 nm and showing the spectra we said the same thing, maybe in a softer way. Anyway, to leave no room for doubt on this point, we now say in the abstract that we have obtained a "better matching to the solar spectrum compared to unmodified norbornadiene" (line 29) and later (lines 155-156) we add “….we obtained only a marginal overlap of the solar spectrum, while for a profitable MOST it would be desirable to reach the visible region of the spectrum”, and in the conclusions, we specify that this matching is "not satisfactory" (line 378) and a few lines below we say again "However, for a productive application of the MOST concept, a better matching of the system to the solar spectrum, possibly achieving photoconversion of norbornadiene in the visible region, must be pursued." (lines 387-389).

Reviewer 3

The Authors have shown that the photoproduct called "tetracycrate" can be converted into a substrate in an exothermic reaction (approx. 66kJ/mol) with the help of a soluble Co(II) porphyrine complex, but they do not suggest how to remove this complex later before the substrate is photochemically reused.

Authors

The idea was to adsorb the catalyst on activated carbon, as already done previously, but we didn't say it. You think about something so much that you end up forgetting to say it, as if it were obvious to everyone.

We have now specified our idea about this by the end of the introduction (from line 115).

Reviewer 3

They did not conduct any studies on whether this immobilized complex is still active or not. Thus, the energy cycle suggested in the article has no evidence for the effectiveness of the concept. I understand that this is preliminary research and perhaps further studies will provide an efficient method of energy accumulation and recovery.

Authors

Having worked with a homogeneous catalyst, it seemed difficult to verify whether it could work in subsequent cycles. We would have to recover it from a solution containing a 20-fold excess of quadricyclane and reuse it. Reflecting on this comment, however, we thought we could carry out a calorimetric measurement using norbornadiene as a titrant and maintaining a fixed quantity of catalyst in the cell. We would have had the limitation due to the volume of the cells but it would have been possible, with the appropriate choice of concentrations, to repeat the catalytic cycle ten times without changing the catalyst, that is, we could have verified the efficiency of the catalyst during 200 turnovers working with 5 % (mol%) of catalyst. We did it and the result was excellent. Not only we verified the value of energy accumulation but we also demonstrated that the catalyst remains 100% active for at least 200 turnovers.

Experimental details of the new measurements have been reported in section 3.3.2. (Isothermal Titration Calorimetry), on lines 326-336, and the results have been presented on lines 203-206.

Reviewer 3

The Authors devoted a lot of space in their work to organic synthesis and here I also have comments and questions. In step 2 of gentle saponification, a solution containing THF was used, for what purpose? The same question applies to stage 4.

Authors

To improve the solubility of reactants in the saponification reactions, mixtures containing THF were used in all cases. There are some references (like: Shi, J., & Niwayama, S. (2018). Practical selective monohydrolysis of bulky symmetric diesters. Tetrahedron Letters, 59(9), 799–802. doi:10.1016/j.tetlet.2017.12.061) indicating that the use of THF as co-solvent improves the yield of the desired monoester.

Reviewer 3

Also, How do you know which ester group has been saponified? I believe that each of the two had the same chances, so the product is probably an equimolar mixture of both possible monoesters, which unfortunately is not visible either in the reaction diagram or in the nomenclature used in the description (line 235), as well as in the description of step 3 (line 250).

Authors

Related to the behavior of ester groups during the saponification, as the referee states, “each of the two had the same chances” since both are equivalents. Thus, the attack of the nucleophile can take place on either of the equivalent carbonyls of the ester functionalities, giving the same mono ester. This is reflected in the nomenclature of the compound, in which the acid group has precedence over the ester group (according to the IUPAC blue book of nomenclature of organic compounds) and thus the carbon of the acid is the C1 and the C bearing the ester group is always the C3.

Reviewer 3

Considering the methods of amide synthesis, why was the transformation process of (1) into (3) not performed in a single step by acting on the diester amine? One explanatory sentence would be useful here.

Authors

Given the high yielding procedure found for the partial hydrolysis of (1) to give (2), the preparation of the amide ester (3) was attempted by amidation of the acid in (2). The two-step sequence cleanly afforded (3) in a combined yield of 85%. However, the partial amidation of (1) was also tried using ethylamine in basic medium. Unfortunately, this one pot procedure gave a lower yield (c.a. 80%) of the amide ester (3), and thus the original two-step procedure was the method of choice.

Reviewer 3

To sum up, I think that the work should be reworded, and the conclusions drawn from it should be more toned down. Once these changes have been made and the supplementary material has actually been made available, I believe it can be published.

Authors

We hope that the changes made to the manuscript meet the requests of this reviewer. We thank her/him again for giving us the opportunity to improve this work.

Reviewer 4 Report

Savastano and co-workers submitted a manuscript titled “A norbornadiene-based molecular system for the storage of solar-thermal energy in aqueous solution: study of the heat-release process triggered by a Co(II)-complex.”  which presented a new water-soluble norbornadiene derivative using in molecular solar thermal energy storage (MOST). This topic is very hot nowadays, and this developed system could be an efficient solution in the energy economy.

The manuscript will be accepted for publication after minor revisions in Exclusive Feature Papers in Inorganic Chemistry 2.0 topic of Molecules.

My comments:

1./ Figure 1 is not high resolution, please change it.

2./ Table 1 in the Conclusions is very confusing and unconventional, please put it into the main text or into the SM.

3./ The caption of Figure 3 has moved, please fix it. (line 130)

Author Response

Reviewer 4

Savastano and co-workers submitted a manuscript titled “A norbornadiene-based molecular system for the storage of solar-thermal energy in aqueous solution: study of the heat-release process triggered by a Co(II)-complex.”  which presented a new water-soluble norbornadiene derivative using in molecular solar thermal energy storage (MOST). This topic is very hot nowadays, and this developed system could be an efficient solution in the energy economy.

Authors

We thanks this reviewer for her/his appreciation of our works and for her/his suggestions.

Reviewer 4

1./ Figure 1 is not high resolution, please change it.

Authors

The quality of the original Figure 1 is good. We will handle the matter with the editorial team to ensure that its final appearance is better than the current one.

Reviewer 4

2./ Table 1 in the Conclusions is very confusing and unconventional, please put it into the main text or into the SM.

Authors

Table 1 has been moved into the text. See 3.3. (Physicochemical Properties of HNBD1/NBD1- and HQC1/QC1-), lines 286-298.

Reviewer 4

3./ The caption of Figure 3 has moved, please fix it. (line 130)

Authors

The position of figure 3 changed depending on the software (program, version) used to read the manuscript. We realized this after submitting the manuscript. Now the problem should no longer exist.

Round 2

Reviewer 2 Report

accept

Author Response

Thank you for your comments.

Reviewer 3 Report

In the submitted corrected and supplemented version of the manuscript, the Authors toned down their conclusions, which I accept. However, they omitted the construction of HNBD1 and HQC1. Let me remind you that after the second stage of synthesis, one of the two ester groups is transformed into a carboxyl one.

The Authors unequivocally indicate then and now which ones, which does not correspond to the chemistry of such compounds. Under the conditions described, when no asymmetric medium was used, two isomeric products remaining in relation to each other as diastereoisomers may be formed in equal quantities, so the product is a racemate.

This fact should have been noted and also included in the chemical name, emphasizing that it is an equimolar mixture of "ethyl 3-(pentyl..... 2,5-diene-2-carboxylate" and "ethyl 2-(pentyl..... 2,5-diene-3-carboxylate".

Similarly, it should be included in the chemical names of tetracyclates. From the point of view of organic chemistry, this is a significant oversight and should be removed before the decision to print the work.

Author Response

We thank this reviewer for her/his additional comments.

For clarity, according to her/his suggestion, we added a sentence (lines 217-218) stating that the final products are obtained as racemic mixtures. However, we would rather leave the chemical names unchanged because we feel that they are correct and give sufficient information. First, when a synthetic procedure does not include any element capable of chiral induction (starting materials, reactants, media), the chirality of the intermediates and products is usually not mentioned, considering that the chirality cannot be created out of achiral elements and processes (in our procedure, we start with a meso compound and use non chiral reagents and media).

On the other hand, according to the IUPAC blue book of nomenclature of organic compounds, the order of precedence for the functionalities involved is carboxylic acid, then ester, then amide. So, the nomenclature proposed by the referee (a mixture of “ethyl 3-(pentyl..... 2,5-diene-2-carboxylate" and "ethyl 2-(pentyl..... 2,5-diene-3-carboxylate”) does not strictly follow the IUPAC rules, since the ester has precedence over the amide and should have the lower locant number (2, and not 3). The referee’s comment may refer to an equimolecular mixture of ethyl (1S,4R)-3-(pentylcarbamoyl)bicyclo[2.2.1]hepta-2,5-diene-2-carboxylate and ethyl (1R,4S)-3-(pentylcarbamoyl)bicyclo[2.2.1]hepta-2,5-diene-2-carboxylate. This is the correct name of the mixture, highlighting the opposite configuration of the chiral carbons of the racemate. The configuration of these carbons did not change along the whole sequence (from a meso compound to a mixture of enantiomers)

For these reasons, we feel that the compounds were correctly named, and there is no need to add the stereochemical descriptors ((1R,4S) and (1S, 4R), or simply +/-).

Also, we disagree with the following referee’s comment: “this is a significant oversight and should be removed before the decision to print the work”.